# New Perspectives of CYP1B1 Inhibitors in the Light of Molecular Studies

Renata Mikstacka [1,*] and Zbigniew Dutkiewicz [2,*]

1    Department of Inorganic and Analytical Chemistry, Collegium Medicum, Nicolaus Copernicus University in Toruń, Dr A. Jurasza 2, 85-089 Bydgoszcz, Poland
2    Department of Chemical Technology of Drugs, Poznań University of Medical Sciences, Grunwaldzka 6, 60-780 Poznań, Poland
*    Correspondence: mikstar@cm.umk.pl (R.M.); zdutkie@ump.edu.pl (Z.D.);
     Tel.: +48-52-585-3912 (R.M.); +48-61-854-6619 (Z.D.)

**Abstract:** Human cytochrome P450 1B1 (CYP1B1) is an extrahepatic heme-containing monooxygenase. CYP1B1 contributes to the oxidative metabolism of xenobiotics, drugs, and endogenous substrates like melatonin, fatty acids, steroid hormones, and retinoids, which are involved in diverse critical cellular functions. CYP1B1 plays an important role in the pathogenesis of cardiovascular diseases, hormone-related cancers and is responsible for anti-cancer drug resistance. Inhibition of CYP1B1 activity is considered as an approach in cancer chemoprevention and cancer chemotherapy. CYP1B1 can activate anti-cancer prodrugs in tumor cells which display overexpression of CYP1B1 in comparison to normal cells. CYP1B1 involvement in carcinogenesis and cancer progression encourages investigation of CYP1B1 interactions with its ligands: substrates and inhibitors. Computational methods, with a simulation of molecular dynamics (MD), allow the observation of molecular interactions at the binding site of CYP1B1, which are essential in relation to the enzyme's functions.

**Keywords:** cytochrome P450 1B1; CYP1B1 inhibitors; cancer chemoprevention and therapy; molecular docking; molecular dynamics simulations

## 1. Introduction

Cytochrome P450 enzymes (CYPs) are hemoproteins representing a superfamily of monooxygenases responsible for the metabolism of endogenous and exogenous substrates. Human cytochrome P450 family 1 (CYP1) consists of three isoforms: CYP1A1, CYP1A2, and CYP1B1. The family 1 member CYP1B1 is an extrahepatic enzyme expressed in hormone-dependent tissues like the breast, prostate, ovaries, pancreas, and testis [1]. CYP1B1 catalyzes oxidation and N- or O-dealkylation of xenobiotics including procarcinogens and environmental pollutants [2], and endogenous substrates such as fatty acids and steroids.

CYP1B1 plays an important role in the pathogenesis of hormone-induced cancers, being responsible for the metabolism of 17-alpha-estradiol (E2) to highly mutagenic and carcinogenic 4-hydroxy-E2 [3,4]. CYP1B1 overexpressed in cancer cells is partly responsible for the resistance to anti-cancer drugs such as tamoxifen, paclitaxel and docetaxel, which are inactivated by this enzyme [5–7]. CYP1B1 is a tumor biomarker because of its overexpression in tumor cells derived from among others: breast, colon, lung skin, brain, and testis [8]. Therefore, the inhibition of CYP1B1 activity is considered a therapeutic target for cancer chemoprevention and chemotherapy. On the other hand, the constitutively overexpressed CYP1B1 in cancer cells and the tumor microenvironment may be used to design anti-cancer prodrugs activated by this enzyme. Another pathway of CYP1B1 studies concerns the relation between mutations of the CYP1B1 gene and the occurrence of eye diseases [9,10] and cardiovascular disorders [11] (Figure 1).

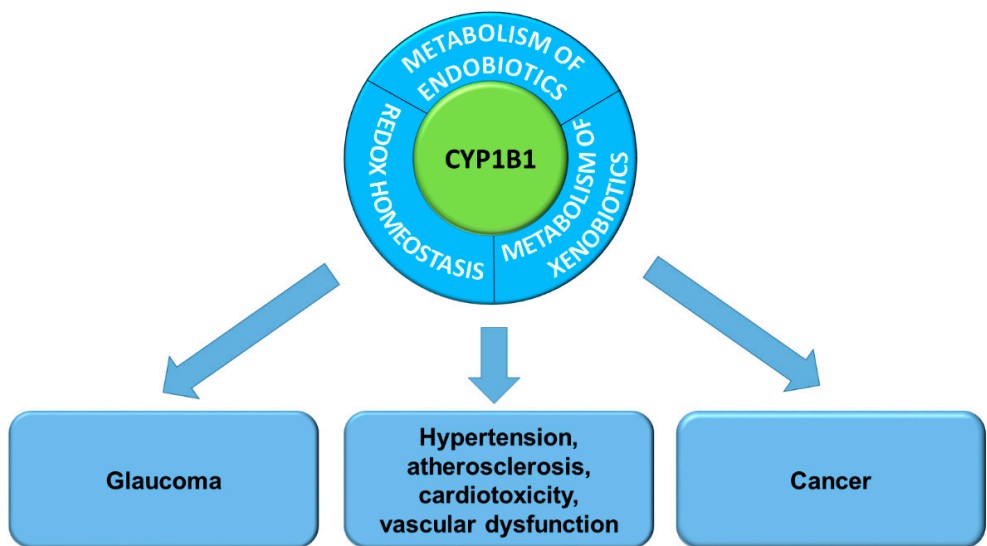

**Figure 1.** Cytochrome P450 1B1 functions and its association with pathogenesis of diseases.

CYP1s are induced by agonists of the aryl hydrocarbon receptor (AhR), including polychlorinated dioxins with strong inducer 2,3,7,8-tetrachlorodibenzo-p-dioxin (TCDD) and polycyclic aromatic hydrocarbons [12,13]. The role of AhR as a transcription factor is not limited to the metabolism of xenobiotics. The AhR transduction pathway is also activated by endogenous compounds, which include the products of metabolism of polyunsaturated fatty acids and heme, and by endobiotics—ligands which are produced in the gastrointestinal tract and skin [14,15].

For a decade, epigenetic factors affecting CYP1B1 gene expression were studied [16]. Methylation of cytosine residues of genomic DNA of the regulatory region(s) of a gene can lead to silencing or diminished expression of the gene. Studies of two cell lines: MCF7 and HepB2, indicated that DNA methylation diminished the expression of the CYP1B1 gene in HepB2 cells [17]. Differential regulation of the expression of CYP1B1 and CYP1B1 genes coding two isoforms of CYP1 was observed in mouse hepatoma and fibroblast cell lines induced by TCDD [18]. In gastric and colorectal cancers, DNA methylation in the promotor region of CYP1B1 was shown [19,20].

This review presents the multiple pharmacological roles of CYP1B1 in a human organism, emphasizing the latest achievements, focusing on the computational studies of the molecular interactions of CYP1B1 with its substrates and inhibitors, explaining CYP1B1 multiple functions at the molecular level.

## 2. The Role of CYP1B1 in the Pathogenesis of Diseases

CYP450 family 1 isozymes were supposed to play an essential role in cancer initiation by transforming procarcinogens into ultimate carcinogens, which react with DNA and form DNA adducts. Recent studies indicate that CYP1B1 contributes to cancer initiation, and to a significant extent, to the following stages of carcinogenesis: promotion and progression, while the primary role of CYP1A1 and CYP1A2 is detoxification of xenobiotics leading to activation of procarcinogens.

In the Human Gene Mutation Database (http://www.hgmd.cf.ac.uk; accessed on 6 May 2021), 240 variants of the CYP1B1 gene are listed. The association of their function with the pathogenesis of diseases in humans has been recently intensively studied [21]. Molecular structures of pathogenic CYP1B1 mutant variants are analyzed in relation to their impaired function with the use of the most advanced bioinformatics tools.

### 2.1. CYP1B1 Contribution in Pathogenesis of Eye Diseases

Glaucoma is a group of eye diseases leading to the irreversible loss of vision. In the development of glaucoma, the CYP1B1 gene is involved: primary congenital glaucoma (PCG)

constitutes 55% of inherited pediatric onset of the disease and is caused predominantly by mutations in the CYP1B1 gene [21].

The studies of CYP1B1 variants and their effect on the risk of eye diseases have been carried out—such studies for PCG and primary open angle glaucoma (POAG) have been carried out since the late nineties (see Section 3.4). In 2006, molecular structures for the eight CYP1B1 mutations occurring only in PCG patients were obtained by comparative modeling and were subjected to MD simulations (30 ns). The impact of studied mutations on the enzyme structure and function was evident [22].

In order to explain the molecular basis of the variable phenotypes resulting from the CYP1B1 gene mutations, the subclones of twenty-three CYP1B1 missense variants occurring in glaucoma patients were examined by measuring in a cell system the dual activity of the enzyme to metabolize both retinol and 17β-estradiol. Most variants linked to POAG showed low steroid metabolism, while null or very high retinol metabolism was observed in variants identified in PCG [23]. In the Pakistani population, identification of five known and three novel variants of CYP1B1 in 14 families with PCG was reported. However, in 22 of the 36 families analyzed (61%), pathogenic alleles were not found [24].

Meta-analysis of seven data sets of Chinese PCG patients revealed L107V and R390H as the most common CYP1B1 mutations, which are localized in close proximity to the active site cavity and hemoglobin binding region. Long-term MD simulations (100 ns) showed the destabilization of the mutant proteins caused the change of the molecular interaction between the enzyme and the substrate [25]. For better reliability of in silico predictions, the authors performed sequence conservation analysis, hydrogen bond network analysis, electrostatic potential analysis, and MD simulation, which all confirmed the impact of CYP1B1 mutations on the function of the enzyme [25].

### 2.2. Redox Homeostasis

The function of CYP1B1 depends on the oxygen supply. Deletion of CYP1B1 gene is associated with increased oxidative stress in the retinal vascular and trabecular meshwork (TM) cells in culture and retinal and TM tissue in vivo [26]. In the absence of *Cyp1b1* in the hepatic sinusoidal endothelial cells prepared from *Cyp1b1−/−* mice, a low level of hepcidin and decreased production of the bone morphogenetic protein (BMP6) were observed [27,28].

Hepcidin is a hormone that regulates the intracellular level of iron by the interaction with the main iron transporter—ferroportin. A low level of hepcidin is associated with the increased iron level and, as a result, increased intracellular oxidative stress. The BMP6 is a critical endogenous regulator of iron metabolism; it activates hepcidin expression and reduces serum iron in mice [29]. Lack of the BMP6 leads to a high intracellular iron level [30].

The role of CYP1B1 in hyperoxia cytotoxicity in human lung endothelial cells in vitro was studied [31]. Human cells overexpressing CYP1B1 were more susceptible to oxygen toxicity, while the use of CYP1B1-siRNA attenuated the toxic effect of hyperoxia. In immortalized lung endothelial cells derived from *Cyp1b1*-null and wild-type mice, *Cyp1b1* expression promoted apoptosis induced by oxidative stress, while in cells derived from *Cyp1b1*-null mice, hyperoxia cytotoxicity was significantly diminished [31]. In studies in vivo, *Cyp1b1−/−* mice lacking the cytochrome P450 1B1 gene were less susceptible to hyperoxic lung injury than C57BL/6 wild type mice [32].

The effect of transcription factor AhR activation on redox homeostasis in basal-like and BRCA1-related breast cancer was reported [33]. In a mouse model of BRCA1-related breast cancer, the levels of reactive oxygen species (ROS) correlated with the expression and activity of the AhR. Chemical or genetic inhibition of the AhR is proposed for maintaining cellular redox homeostasis, modulation of the tumor promoting the microenvironment, and overcoming the resistance of human breast cancer cells to erlotinib used in anti-cancer therapy.

## 2.3. CYP1B1 in Cardiovascular Diseases

In the last decade, the implication of CYP1B1 in cardiometabolic diseases has been recognized and intensively studied [34]. CYP1B1 is responsible for the cardiovascular side effects of anti-cancer treatment. Cardiovascular disease is the second cause of mortality in the group of cancer survivors, after secondary malignancies. Inhibition of the CYP1B1 can be a promising therapeutic strategy that has the potential to prevent firstly: chemo- and radio-resistance to cancer therapy and secondly, treatment-induced cardiovascular complications without reducing their anti-cancer effects [11]. Most chemotherapeutic agents like doxorubicin, dasatinib, sunitinib, and docetaxel induce CYP1B1 gene expression in cardiac-derived cells in vitro and in animal models in vivo [11], leading to the resistance to cancer chemotherapy. Increased CYP1B1 expression is associated with cardiac hypertrophy [35–38], hypertension [39–43], atherosclerosis [44], cardiotoxicity, and vascular dysfunction. 6β-Hydroxytestosterone, a metabolite of testosterone generated by CYP1B1, contributes to vascular changes in angiotensin II-induced hypertension in male mice [45].

Efficacy of CYP1B1 inhibition in protection against chemotherapy-induced cardiotoxicity and in the prevention of chemo- and radio-resistance has been demonstrated in numerous preclinical studies. Phytochemicals such as quercetin, apigenin, and other polyphenols exhibit protective activity against cancer chemotherapeutics' side effects [11]. However, the activity of these agents is directed to many other targets and is also responsible for drug interactions. Their antioxidant activity may impair ROS-dependent cancer therapy.

2,4,3′,5′-tetramethoxy-trans-stilbene (TMS), a synthetic derivative of resveratrol as a selective CYP1B1 inhibitor protected from chronic DOX-induced cardiotoxicity in rats in vivo and RL-1 cardiomyocyte-like cells in vitro through a mid-chain hydroxyeicosatetraenoic acid (HETE)-dependent mechanism. Overexpression of CYP1B1 significantly induced cellular hypertrophy and the production of mid-chain HETE metabolites. At the same time, inhibition of CYP1B1 with the use of 2,4,3′,4′-tetramethoxy-*trans*-stilbene (TMS) or CYP1B1-siRNA significantly attenuated isoproterenol-induced hypertrophy [37].

## 2.4. Metabolic Diseases

CYP1B1 is considered a therapeutic target for the treatment of metabolic diseases [34]. CYP1B1 metabolizes endogenous substrates, including steroid hormones, fatty acids, melatonin, vitamins, and retinoids. The disorders of endogenous metabolism lead to metabolic diseases that may result in obesity, atherosclerosis, hypertension, and cancer. In the studies on *Cyp1b1*-knockout mice, CYP1B1 deficiency reduced obesity induced by high fat diet—when compared with their wild-type counterparts, and improved glucose tolerance [46]. *Cyp1b1* deletion caused suppression of peroxisome proliferator-activated receptor γ (PPAR γ) and the genes regulated by PPARγ [47]. PPARs are nuclear receptors involved in the homeostasis of lipids and glucose. To summarize, modulation of CYP1B1 activity attenuates the processes that lead to obesity, hypertension, and atherosclerosis. CYP1B1 is expressed in vascular smooth muscle cells (VSMCs) and, to a lesser extent, in endothelial cells [48] and is involved in the development of hypertension via the regulation of arachidonic acid metabolism. Inhibition of CYP1B1 activity by TMS resulted in decreased ROS production and extracellular signal-regulated kinase $\frac{1}{2}$ and p38 mitogen-activated protein kinase activity and significantly alleviated hypertension [49].

## 2.5. The Role of CYP1B1 in Cancer Chemoprevention and Therapy

Regarding its functions in cellular metabolism, CYP1B1 is considered as a target in cancer chemoprevention and therapy (Figure 2). Cancer chemoprevention is the strategy to prevent or delay the initiation, promotion, or progression of cancer disease with the use of natural compounds or their synthetic derivatives. Progress of cancer depends on the activation of pathways leading to inflammation and angiogenesis [50–52]. Many phytochemicals exert anti-inflammatory and anti-angiogenic activities [53–55]. Most of these compounds, like flavonoids, furanocoumarins, chalcones, stilbenoids, anthraquinones, and alkaloids, are also potent and specific inhibitors of CYP1B1 activity [56,57]. Thus, the association of

CYP1B1 with the inflammatory process and angiogenesis is suggested. Therefore, the use of CYP1B1 inhibitors targeting both cancer cells and their microenvironment, including neutrophils, macrophages, lymphocytes, and fibroblasts, is supposed to be a promising strategy in chemoprevention [58]. CYP1B1 may contribute to the cytotoxicity of natural compounds against cancer cells and cancer stem cells resistant to chemotherapy [59,60].

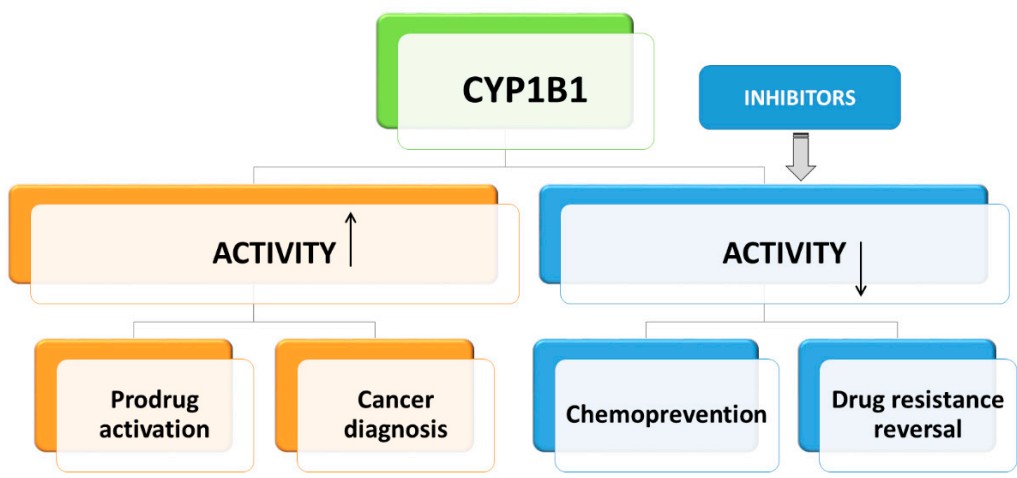

**Figure 2.** The role of CYP1B1 in cancer chemoprevention and therapy.

The mechanism of CYP1B1 inhibition is analyzed with the use of computational methods to obtain safe and more efficient chemopreventive agents [61]. Since the 1990s, natural stilbenoids, *trans*-resveratrol (RESV), pterostilbene, piceatannol, and synthetic RESV analogs have been extensively studied in relation to chemoprevention [62] and with their interaction with CYP1s. Synthetic analogs of resveratrol were screened for their inhibitory activity towards cytochrome P450 family 1. TMS synthesized two decades ago is still supposed to be one of the most potent inhibitors of CYP1B1 activity. CYP1B1's interactions with inhibitors and their effect on conformational changes in the enzyme structure were studied with a series of 3,4-dimethoxy-*trans*-stilbene derivatives docked to the crystal structure of human CYP1B1 [63].

One of the most critical problems in cancer therapy is overcoming drug resistance. Flutamide is an example of an anti-cancer chemotherapeutics metabolized by CYP1B1 to a 2-hydroxy derivative that does not exert anti-cancer activity [6]. More recently, the properties of flutamide as an agonist of AhR were shown [64]. Increased expression of CYP1B1 results in decreased flutamide efficacy as an anti-cancer agent. CYP1B1 can be a potential tumor biomarker and a target for anti-cancer therapy in renal cell carcinoma (RCC). Expression of CYP1B1 was determined in RCC cell lines: Caki-1 and 769-P, and tissue microarrays of 96 RCC and 25 normal tissues. The gene CYP1B1 in Caki-1 and 769-P cells was silenced by RNA interference, and functional analyses were performed to determine the biological significance of CYP1B1 in RCC progression. CYP1B1 promoted RCC development by inducing the cell division cycle 20 homolog ($CDC_{20}$) expression and inhibiting apoptosis through the down-regulation of the death-associated protein kinase-1 (DAPK1) [65].

CYP1B1 is overexpressed in primary and metastatic loci of epithelial ovarian cancers. Treatment with α-naphthoflavone reduces paclitaxel resistance and enhances the sensitivity of ovary cells to paclitaxel in vitro and in the xenograft model of nude mice [7]. A new highly selective and extremely potent α-naphthoflavone derivative ($IC_{50}$ = 0.043 nM) was synthesized. It overcomes docetaxel-resistance caused by the overexpressed CYP1B1 in MCF-7/1B1 cells [66].

To summarize, co-administration of the anti-cancer agents and CYP1B1 inhibitors is suggested to decrease drug resistance and ameliorate the outcome of anti-cancer therapy.

Investigation of the molecular mechanism of CYP1B1 contribution in oncogenesis revealed that CYP1B1 promotes cell proliferation and metastasis by inducing epithelial-mesenchymal transition (EMT) and Wnt/β-catenin signaling via Sp1 (a transcription factor involved in cell growth and metastasis) induction [67]. CYP1B1 induces the development and progression of prostate cancer cells by modulating caspase 1 (CASP1) expression. An inverse correlation between CYP1B1 and CASP1 in human prostate cancer samples was shown. In vivo lentivirus-delivered short hairpin (sh) RNA specific for CYP1B1 was used to reduce the tumor progression in the xenograft mouse model. The results support the idea that attenuation of CYP1B1 may be effective in the treatment of prostate cancer [68].

Regarding the high level of CYP1B1 activity in cancer cells, the design of prodrugs that could be activated by CYP1B1 directly in a tumor is still a developing area of study. In vitro models were proposed to screen CYP1B1- targeted anti-cancer prodrugs. In KLE cells (human endometrial carcinoma cell line), which overexpress CYP1B1 activity, the increased cytotoxicity of studied prodrugs was observed in comparison to cells with CYP1B1 inhibited by αnaphthoflavone (ANF) that is a potent inhibitor of CYP1 isozymes [69]. Recently discovered DMAKO-20 (5,8-dimethyl alkannin oxime derivative) is an anti-cancer prodrug activated by CYP1B1, cytotoxic to cancer cells in vitro and in vivo, while it is nontoxic to the human normal cells [70].

A new approach in tumor diagnosis is based on CYP1B1 expressed in tumor cells. The derivative of ANF, 3-hydroxy-3′-fluoro-6,7,10-trimethoxyANF, was earlier characterized as a very potent CYP1B1 inhibitor with $IC_{50}$ equal to 0.043 nM [66]. This compound modified with a linker still potently inhibited CYP1B1 with $IC_{50}$ equal to 4 nm. Near-infrared fluorescent dye was attached to ANF derivative and used as CYP1B1 targeted imaging probes in HCT-15-tumor bearing mice [71]. This non-invasive method of imaging cancer cells is a promising approach in tumor diagnosis.

## 3. CYP1B1 Molecular Structure Studies

### 3.1. Family CYP1 Crystal Structures

The insight into molecular structures of CYP1 isozymes provides much valuable information due to the development of computational methods: molecular docking and molecular dynamics simulations. All members of the family CYP1 have flat and planar active sites. CYP1B1 protein consists of 543 amino acids and has a slightly smaller active site than CYP1A1. It has low amino acid sequence identity with CYP1A1 (38%) and CYP1A2 (37%), but it exhibits similar substrate specificity [72–74]. Until now, a single molecular structure of human CYP1A2, four human CYP1A1 structures, and two human CYP1B1 structures differing in the bound ligands are available. Recently, two new structures of CYP1B1 were deposited in Protein Data Base (unpublished data). Besides, two ancestral CYP1B1 structures were generated and characterized (Table 1).

**Table 1.** Crystal structures of CYP1 isozymes: CYP1A1, CYP1A2 and CYP1B1 deposited in Protein Data Base.

| CYP | PDB Id | Ligand | Reference |
| --- | --- | --- | --- |
| CYP1A1 | 4I8V | α-naphthoflavone | [74] |
| | 6DWM | bergamottin | [75] |
| | 6DWN | erlotinib | [75] |
| | 6O5Y | Pim kinase inhibitor GDC-0339 | [76] |
| | 6UDM | Duocarmycin Prodrug (S) ICT-2726 | to be published |
| | 6UDL | Duocarmycin Prodrug (S) ICT-2700 | to be published |
| CYP1A2 | 2HI4 | α-naphthoflavone | [77] |
| CYP1B1 | 3PM0 | α-naphthoflavone | [73] |
| | 6IQ5 | inhibitor having azide group | [78] |
| Ancestral CYP1B1 | 6OYU | α-naphthoflavone | [79] |
| | 6OYV | estradiol | [79] |

According to the size and shape of the binding site cavity, relatively small and planar molecules as polycyclic aromatic hydrocarbons (PAH), stilbenes, flavonoids, and anthraquinones are substrates/inhibitors of CYPs family1 [80,81]. Structure-based drug design requires detailed knowledge of active-site interactions with potential ligands. In studies of substrate-enzyme interactions, computational methods, particularly long-term molecular dynamics simulations, that identify the subtle changes of molecular conformations provided the elucidation of how non-planar substrates might be accommodated in small CYP's binding sites. Currently used bioinformatics tools are briefly discussed in Section 3.5. The studies on CYP1A1 molecular structures revealed that the accommodation of the substrates with higher molecular weight and non-planar structure in the enzyme binding site is possible due to conformational changes of the side chains in the enzyme cavity [75,76].

Crystallographic CYP1A1 structure with bound ANF (PDB: 4I8V) was characterized in 2013 by Walsh and coworkers [74]. For CYP1A1, two new molecular structures were solved with a furanocoumarin bergamottin (PDB: 6DWM) and tyrosine kinase inhibitor erlotinib (PDB: 6DWN) as ligands [75]. All three structures show a small planar active site. In the case of the crystallographic structure of CYP1A1 with the non-planar substrate, Pim kinase inhibitor GDC-0339 (PDB: 6O5Y) molecular docking revealed the significant changes of shape and size of CYP1A1 cavity as a result of interactions with the ligand [76]. The molecular CYP1A1 structure was controlled by residues in the active site roof with major changes in the conformation of the F helix break and relocation of Phe224 from the active site to the protein surface [76]. The structural rearrangements were observed in the CYP1A1 binding site when diverse ligands were cocrystallized with the enzyme [76]. Molecular dynamics allow discerning the conformational changes in the CYP1A1 cavity, which facilitate the accommodation of the compounds of the studied series of stilbene derivatives in the binding site. These changes included the repositioning of side chains of amino acids located in the roof of the CYP1A1 active site [76].

### 3.2. CYP1B1 Crystal Structure Studies

Since 2011, a single crystal structure of CYP1B1 with ANF as the bound ligand (PDB id: 3PM0) determined by X-ray crystallography to 2.7 resolution Å was available [73]. Previously, the interactions of natural and synthetic inhibitors with CYP1B1 catalytic site were estimated using CYP1A2 based homologous structures [61]. Both enzyme structures are generated with ANF as a ligand; however, ANF binds in a different orientation in P450 1B1 from that observed for 1A2 (Figure 3). A distortion of helix F places Phe231 in CYP1B1 and Phe226 in CYP1A2 in similar positions for π-π stacking with ANF. Hydrophobic interactions between ANF and amino acid residues are the predominant interactions that contribute to binding affinity.

More recently, the new structure of CYP1B1 with inhibitor having azide group as a ligand (PDB id: 6IQ5) was crystallized [78]. In this research, ANF was used as a lead compound to design efficient CYP1B1 inhibitors with better solubility and inhibitory potency. Synthesized benzo[*h*]chromone with linked cyclohexanyl B-ring exerted a strong CYP1B1 inhibitory activity with potency more than 2-times stronger than that of ANF. Moreover, the introduction of azide substituent caused a 120-fold increase of selectivity against CYP1B1 compared to CYP1A1. X ray crystal structure allowed to confirm the location of the inhibitor molecule in the CYP1B1 cavity, while the positioning of terminal nitrogen atom could not be determined due to the low resolution. The authors performed docking analysis using Surflex Dock program on Sybyl X2 software and found that the azide group was located in the hydrophobic pocket constructed by Thr334, Val395, and Thr510. The docking results exhibited the role of hydrophobic interactions affecting the binding affinity and inhibitory potency of the ANF derivative (Figure 4).

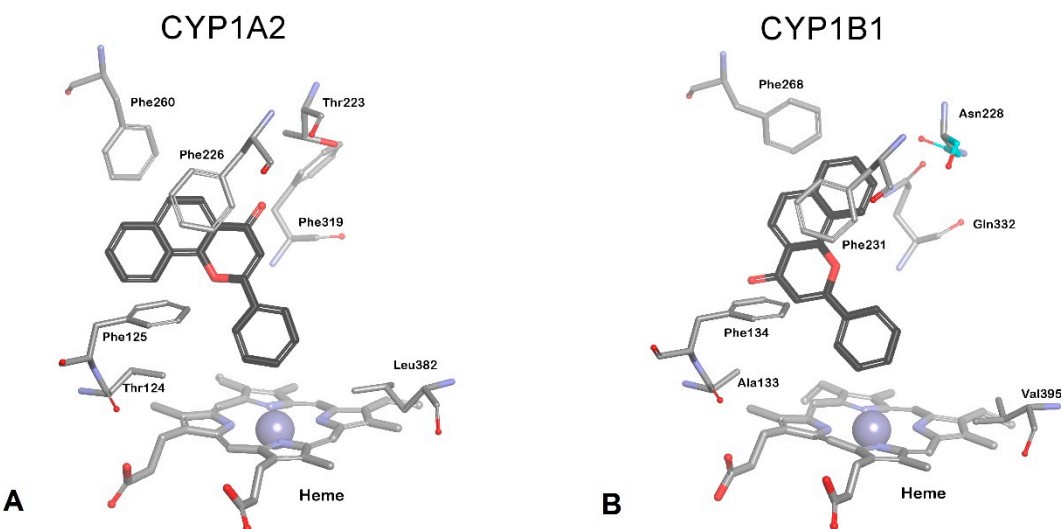

**Figure 3.** Orientation of ANF in: (**A**) CYP1A2 (PDB id: 2hi4) and (**B**) CYP1B1 (PDB id: 3pm0). ANF (rendered with *black* carbon atoms), heme, and residues are presented as stick models in atom type.

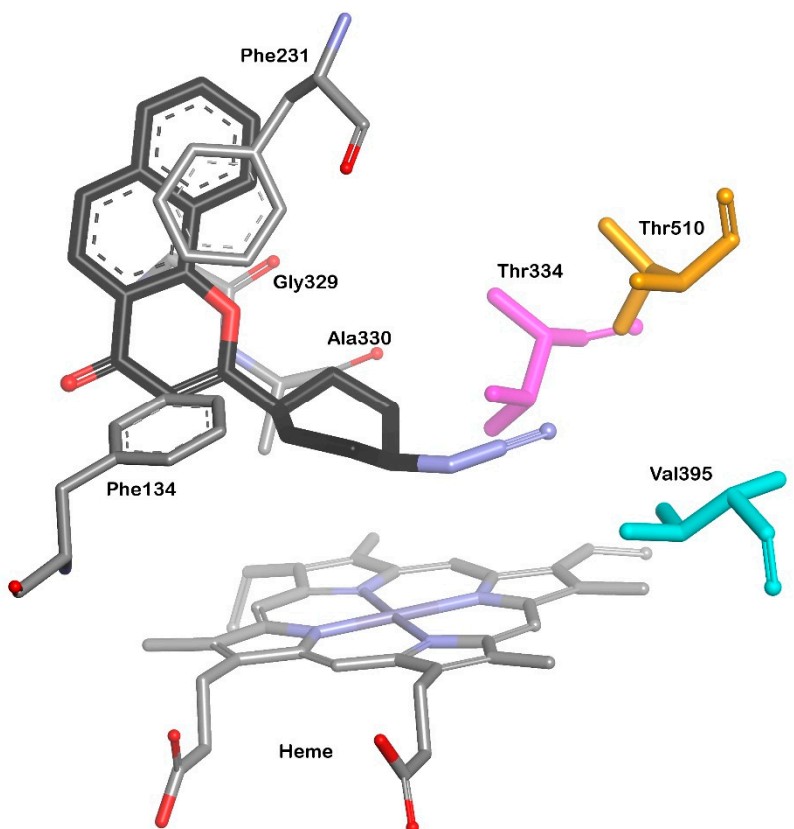

**Figure 4.** The orientation of inhibitor with azide group (benzo[*h*]chromone derivative) in CYP1B1 binding site. Inhibitor molecule (black carbon atoms); heme, Phe134, Phe231, Gly329, and Ala330 residues are presented as stick models in atom type. Thr334 (magenta), Val395 (cyan) and Thr510 (orange) as stick models.

The ability of human P450s to catalyze a plethora of chemical reactions makes these enzymes particularly valuable for the commercial biosynthesis of important molecules. However, the use of cytochromes P450 in biotechnology for the synthesis of chemicals, including final drugs and drug metabolites, requires higher thermostability than

that of human cytochromes. The ancestral mammalian cytochrome P450 subfamily 1B (N98_CYP1B1_Mammal) was reconstructed to investigate its thermostability, substrate selectivity, and metabolism of representative substrates. The structure of ancestral CYP1B1 was determined with ANF and 17-estradiol in the binding site [79]. Comparing the human CYP1B1 (PDB id: 3PM0) and N98_CYP1B1_Mammal (PDB id: 6OYU) complexed with ANF enabled the identification of the similarities and differences in their structures. First of all, predicted ancestral CYP1B1 has maintained a canonical cytochrome P450 fold, and the sequence differences among the variants were found far from the active site. The difference that was supposed to have critical implication to the enzyme catalytic activity concerned the F helix exhibiting characteristic break in the CYP1 isozymes. In the ancestor of CYP1B1, this break is not present; instead, the F helix is extended by eight residues comparing with the human CYP1B1. The elongation of the F helix results in the appearance of a wide channel from the active site to the protein surface, what in turn leads to the binding as many as four ANF molecules (Figure 5). This open conformation of the enzyme is also preserved in the structure of the ancestral CYP1B1 with only one molecule of 17-estradiol as a ligand. It is assumed that electrostatic and aromatic interactions between distinct secondary structure elements in the ancestral form contribute to its higher thermostability than that of the extant human CYP1B1 [79].

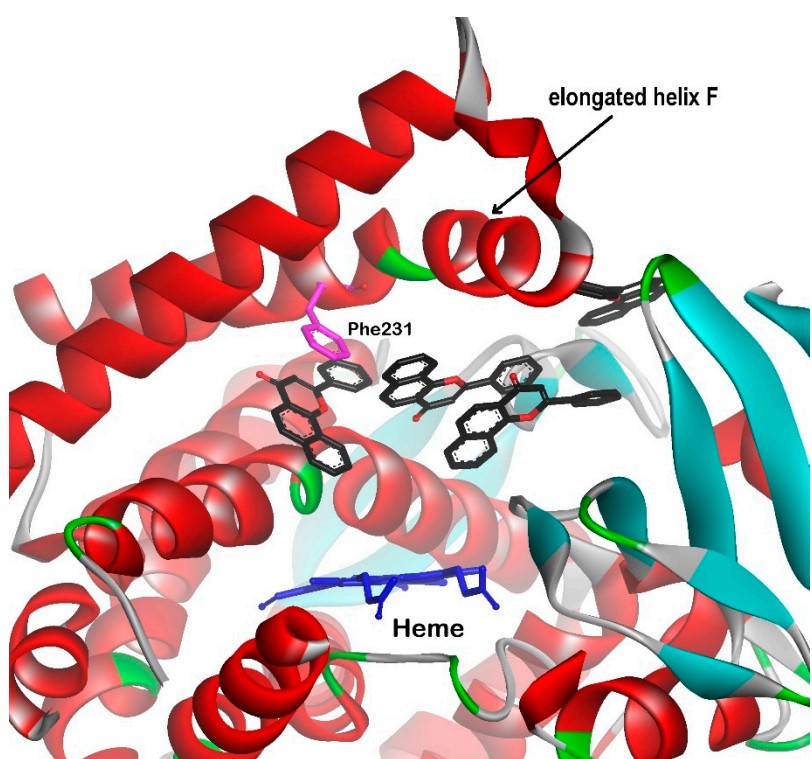

**Figure 5.** Four molecules of ANF bound to ancestral CYP1B1 (PDB id: 6OYU). ANF (black carbon atoms), heme (dark blue), and Phe231 (magenta) presented as stick models.

Crystallography and molecular dynamics (MD) simulation studies on the cytochrome P450 structures of different isozymes demonstrated the importance of gating amino acid residues associated with the opening and closing of the ligand tunnels. Lipophilicity of a ligand is another factor influencing its penetration into the enzyme active site. Cytochromes P450 are membrane bound enzymes, and the ligand can enter the active site from the lipophilic membrane and the aqueous environment. MD simulations allow the discovery of multiple ligand tunnels in cytochrome crystal structures. CYP1B1 structure (3PM0), has the most enzyme crystal structures available in the Protein Data Base, was generated in the closed conformation. The opening and closing motions of most ligand tunnels are associated with conformational changes of the flexible B–C and F–G blocks

of the cytochromes. Understanding the gating mechanism can help analyze the catalytic mechanism and substrate selectivity of CYPs [82].

### 3.3. Design of Selective CYP1B1 Inhibitors

Inhibitors are chemical tools used in the studies of the molecular structure of enzymes. In turn, the knowledge of enzyme structures facilitates the studies on its function and allows the design of new molecules with stronger affinity to the binding site. In structure-based drug design (SBDD), many methods can be used to calculate the binding affinity of the molecules for their biological target prior to their synthesis [83]. Computational methods in the search for new efficient CYP1B1 inhibitors are widely used, highlighting the mechanism of enzyme inhibition and indicating possible structural modifications of known inhibitors and drug candidates, thereby improving inhibitory properties. A structure–activity relationship is a crucial issue in drug design development. Strategies in search of target-specific inhibitors were classified by Zhan and coworkers as target-guided approaches and derivatization approaches based on the known substrates and inhibitors [84]. Taking into account the design of effective inhibitors of individual enzyme isoforms, an additional feature to be met is the selectivity of their action [85]. The shape of the enzyme cavity, electrostatic properties of amino acids on its surface, the flexibility of the active site, and the location of water molecules bound in the active site must be taken into account in the process of the design of selective and potent enzyme inhibitors [86].

Stilbenoids and flavonoids (Figure 6) are two classes of compounds for which interactions with CYP1 isozymes were studied extensively for two decades. Stilbene and flavonoid derivatives were designed, synthesized, and evaluated as substrates and inhibitors of CYP1 isozymes with special regard to their molecular interactions in the enzyme binding sites. The high potency of *trans*-stilbene methoxy derivatives as selective inhibitors of CYP1B1 activity was analyzed with the use of computational methods, among others stilbene derivatives: 2,4,3′,5′-tetramethoxy-*trans*-stilbene (TMS) and 2,2′,4,6′-tetramethoxy-trans-stilbene [87–89]. In our studies, 2,3′,4′-trimethoxy-*trans*-stilbene was the most effective and selective CYP1B1 inhibitor among the studied series of poly-methoxystilbenes [63]. Molecular docking of 2,3′,4′-trimethoxy-*trans*-stilbene to the enzyme binding site with the use of the LigandFit procedure revealed that van der Waals forces were mostly responsible for the high affinity of this compound to the CYP1B1 binding site. In the energetically favorable orientation of 2,3′,4′-trimethoxy-*trans*-stilbene, 3′,4′-dimethoxy motif was directed to the heme, and π-π stacking of the phenyl ring with Phe231 was formed.

To explain the selectivity of 2,3′,4′-trimethoxy-trans-stilbene towards CYP1B1, its interactions at the CYP1B1 binding site were compared to 2,4′-dimethoxy-*trans*-stilbene, which inhibited CYP1B1 activity about 20 times less. Energetically favored poses of 2,4′-dimethoxy-*trans*-stilbene were directed to the heme with 4′-methoxyphenyl. The oxygen atom in position 2 of 2,4′-dimethoxy-*trans*-stilbene formed the hydrogen bond with the $NH_2$ group of Gln332, whereas 2,3′,4′-trimethoxy-*trans*-stilbene interacted with Gln332 by means of π electrons of the stilbene phenyl ring. Additionally, π electrons of both compounds interacted with the peptide bond Gly329/Ala330 (π-amide stacking). For 2,4′-dimethoxy-*trans*-stilbene docked to the CYP1B1 binding site, the weak hydrogen bond between Asn228 and the polarized $CH_3$ group (C-H—O=C) in position 2 was observed. For 2,3′,4′-trimethoxy-*trans*-stilbene oriented with 3,4-substituted phenyl to the heme in the CYP1B1 binding site, T-shaped π-π interaction occurred [90].

In a series of *polymethoxy-trans*-stilbenes, tetramethoxy and pentamethoxy derivatives displayed potent inhibitory activity toward CYP1A1 and CYP1B1 with $IC_{50} < 1$ μM, while the derivatives possessing methoxy substituents in positions 4 and 4′, which made the molecules longer, did not fit to the CYP1A2 binding site [63]. This confirms the findings of Liu et al. concerning the size and shape of CYP1cavities. CYP1A1 and CYP1B1 have narrow and long cavities compared to the CYP1A2 binding site, which can accommodate triangular molecules, showing a planar trigonal structure with side lengths 9.3 Å, 8.7 Å, and 7.2 Å, respectively [91].

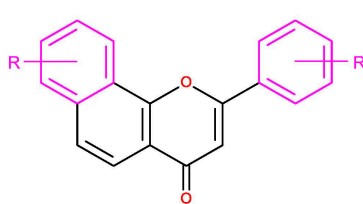

(**a**) stilbenoids

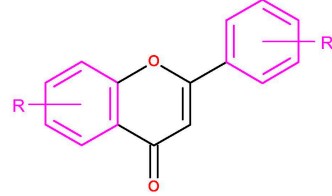

(**b**) flavonoids

(**c**) naphtoflavonoids

(**d**) benzo[h]chromone
derivatives

(**e**) steroids (estrane derivatives)

(**f**) benzochalcone derivatives
with modified B ring

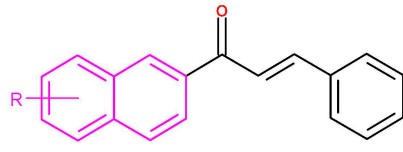

(**g**) benzochalcone derivatives
with substituted naphthalene-
ring

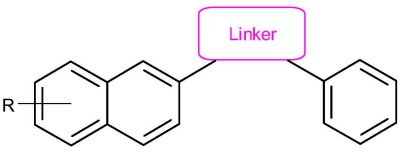

(**h**) benzochalcone derivatives
with modified linker

(**i**) carvedilol

**Figure 6.** Core structures (scaffolds) used in the design of selective CYP1B1 inhibitors: (**a**) *trans*-stilbene derivatives; (**b**) flavone derivatives substituted in A and B rings, (**c**) α-naphtoflavone derivatives with B and D ring modifications; (**d**) benzo[*h*]chromone derivatives with various B-ring (X) structures; (**e**) pyridine-, 3-thio- and 2-aryl-estrane derivatives; (**f**) benzochalcone derivatives with modified B ring; (**g**) naphthalene ring-substituted benzochalcones; (**h**) linker modified benzochalcones; (**i**) carvedilol as a novel CYP1B1 inhibitor found in structure-based virtual screening.

More recently, the selectivity of stilbenoid inhibitors against CYP1 isoforms was studied with integrated fit docking and molecular dynamics simulation. The potency

of CYP1 inhibitors was correlated with π-π stacking interactions with the phenylalanine residues in the F helix. Modification of 2,4,3',5'-tetramethoxy-trans-stilbene with thiophene and imidazole rings significantly improved the selectivity of the modified inhibitor against CYP1A2, while the potency against CYP1B1 did not change [85].

The effect of natural and synthetic flavonoids and naphthoflavonoids (Figure 6) on CYP1 activity was an objective of extended structure–activity studies [92]. Among a series of 33 flavonoids, 3,5,7-trihydroxyflavone (galangin) inhibited CYP1B1 activity with the IC50 value of 3 nM. Six flavone derivatives were docked to CYP1B1, indicating the occurrence of more than one binding site in the CYP1B1 cavity [93]. A and/or C rings were found near SRS4 and SRS5 and far from SRS1, SRS2, and SRS3. In these studies, the authors generated CYP1B1 structure by homology modeling based on the crystal structure of CYP1A2.

ANF is the best known strong inhibitor of CYP1s; however, this compound is weakly soluble in water, and it does not exert selectivity toward CYP1B1. Choosing ANF as a lead compound, its derivatives differing in the substituents and their positions were designed and synthesized. Cui and Dong with coworkers [66,94] set out some structural features determining the inhibitory potency of designed molecules. In their studies of a series of ANF derivatives, very potent CYP1B1 inhibitors were found, indicating the indispensable role of the keto group at the C-ring and the significant increase of inhibitory potency when a compound possessed a halogen atom at the 3' position on the B-ring. ANF derivative, 3-hydroxy-3'-fluoro-6,7,10-trimethoxy-7,8-benzoflavone exhibited the highest potency against CYP1B1 ever reported with IC$_{50}$ equal to 0.043 nM [65]. Equally high inhibitory potency with IC$_{50}$ below 1 nM was found for B ring modified ANF derivatives [95]. The compounds possessing chloro- and trifluoromethyl groups substituted at position 3 of the B ring were 10-fold more potent than the parent compound. Molecular docking of 3-chloro-ANF derivative revealed the π-π stacking interaction with Phe231 and three hydrogen bonds to the amino and hydroxyl groups of Ser131 and to the carbonyl group of Asp326 with distances of 3.5, 2.1, and 2.8 Å, respectively. In the CYP1B1 active site, ANF derivatives adopt a different pose than the ANF binding pose. As in cases of other inhibitors, the affinity to CYP1B1 was increased by hydrophobic contacts with Ile399, Leu264, Phe268, Phe120, Ala330, and Ala121 [95]. Benzochalcones (Figure 6) was another series of synthesized compounds related to ANF, exhibiting a nanomolar inhibitory potency toward CYP1B1 and the cytotoxic activity against drug-resistant cancer cells [96].

In turn, two heterocycle-containing ANF derivatives are noteworthy as very potent CYP1B1 inhibitors: a fluorine-containing pyridin-2-yl substituted 6,7,10-trimethoxy-ANF (IC50 value of 0.07 nM) and a pyridin-3-yl substituted 6,7,10-trimethoxy-ANF with the 4-NH$_2$ group at the ring B (IC50 value of 0.98 nM), whose solubility in water is much better than that of ANF [94].

The design of CYP1B1 inhibitors may be based on their substrates as lead compounds. Four series of A-ring substituted steroids (Figure 6) were synthesized, and their inhibitory activity against CYP1B1 was estimated. 2-(4-Fluorophenyl)-E2 was the best CYP1B1 inhibitor with IC$_{50}$ = 0.24 μM. Docking results showed that this new estrane derivative fits better into the CYP1B1 binding site than that of CYP1A1 because its A-ring produces more π-interactions in the CYP1B1 cavity in comparison with CYP1A1. These included π−π T-shaped with Phe-231 and π−σ with Ala-330. Asn228 residue formed a hydrogen bond with the 17β-OH [97].

Another way to search for efficient and selective CYP1B1 inhibitors is the structure-based virtual screening of approved drugs registered in the FDA database, which Wang and his group applied. The screening results were correlated with experimental determination of inhibitory activity, indicating beta-blocker carvedilol (Figure 6) as a promising CYP1B1 inhibitor [98]. Carvedilol, a drug used in the treatment of high blood pressure and heart failure, was docked to the X-ray crystal structure of the enzyme (PDB code: 6IQ5). Its binding pose in the CYP1B1 cavity was stabilized by π-π stacking interaction formed by a 9H-carbazole group of carvedilol with Phe231. As in other cases of CYP1B1 inhibitors,

hydrophobic interactions formed by the methoxy-substituted phenyl with heme and surrounding residues decided on the affinity of carvedilol to the CYP1B1 binding site [98].

### 3.4. Genetic Variability

CYP1B1 involvement in cancer therapy requires extensive knowledge of the CYP1B1 molecular structure, which determines its interactions with the ligands. Polymorphism of CYP1B1 is an objective of research concerning the increased risk of cancer [99,100]. Genetic variability of CYP1B1 was studied since 1997 when Stoilov and coworkers identified three truncated mutations of CYP1B1 as the cause of primary congenital glaucoma [9,101]. Pharmacological analyses of individual CYP1B1 variants indicated univocally on the association of CYP1B1 mutations with eye diseases [102–105].

The structures of CYP1B1 variants responsible for primary congenital glaucoma (PCG) and primary open angle glaucoma (POAG) were earlier generated using bacterial CYP102 or human CYP2C9 as a template [9,22,101]. More recently, structural determinants of the CYP1B1 variants associated with PCG and POAG in humans were investigated with the use of crystal structure generated with ANF as a ligand. In 2016, extensive normal mode analysis and MD simulations performed by Banerjee and coworkers [23] enabled the observations of possible effects of mutations on the CYP1B1 structure. Altered structural flexibility within the B-C and F-G loop region and altered tunnel properties were observed in a variant with changed metabolizing activity of 17-β-estradiol (E2) relative to the normal level [23].

Other authors indicate that the I helix plays an essential role in substrate binding, imposing its orientation in the binding cavity and consequently deciding on the site of metabolism. Valine at position 395 decides the stereochemistry of E2 hydroxylation. Wild type CYP1B1 predominantly metabolizes E2 to 4-OH-E2, while 2-OH-E2 is a product of CYP1A1 having leucine at position 395. Consistently, CYP1B1 mutant V395L, having valine replaced with leucine, also catalyzes hydroxylation at the 2-position due to the change in substrate orientation in relation to ferryl oxygen [106].

CYP1B1 variants are also investigated with regard to the metabolism of environmental pollutants [2]. Both allelic variants CYP1B1.1 and 1B1.3 preferentially oxidized 1-chloropyrene, a chlorinated polycyclic hydrocarbon, at the 6-position, while CYP1A1 catalyzed hydroxylation at the 6- and 8-positions more actively than that at the 3-position [107]. The orientation of 1-chloropyrene in the CYP1B1 binding site (3PM0) was determined with position 6 directed to the heme in the distance of 5.71 Å. However, the CYP1B1 variants did not differ in activity toward 1-chloropyrene. The structure of the CYP1B1 active site was compared with that of CYP2A13, displaying fargoing similarities, and both enzymes also showed overlapping substrate specificity [108].

The analysis of interactions between polymorphic forms of CYP1B1 and substrates/inhibitors is explored concerning the design of specifically targeted anti-cancer agents. The hypothesis that CYP1B1 variants coding impaired protein with reduced enzyme function is responsible for better response to anti-cancer treatment still needs to be confirmed. In the Chinese population, women with the homozygous variant genotype for CYP1B1 (single nucleotide polymorphism rs1056836) exhibited a significantly reduced risk of breast cancer; however, homozygous variant genes for GSTP1 and COMT were associated with an increased risk of developing breast cancer [109]. Several reports have shown that the CYP1B1.3 genetic variant (L432V) is associated with an increased risk of prostate, colorectal, and urinary system cancers [68,110–113].

### 3.5. Characteristics of in Silico Methods

The quality and reliability of the results obtained in molecular docking and molecular dynamics (MD) simulations largely depend on the models, approximations, and computational methods used in the experiments. Docking, used as a virtual screening tool to assess the affinity of ligands, uses fast, yet approximate methods, i.e., scoring functions (SFs). Commonly used SFs belong to four groups: force field-based, empirical, knowledge-based,

and relatively new, machine-learning-based [114–117]. Better at distinguishing active from inactive molecules are scoring functions based on quantum mechanics, which give a much lower number of false positives, although at the cost of increased computation time by one to two orders of magnitude [118–122]. For ligand design, where individual molecules or a series of congeneric ligands are docked, the affinity assessment helps determine the most likely orientation (pose) of the ligand, which in turn determines the possibility of specific ligand-protein interactions [63,66,85,90,93–97].

The binding of a ligand to a protein usually changes the conformation of both the ligand and the protein. Thus, it is also important for flexible ligands to properly sample their conformational space during docking [116]. The more significant challenge is to account for the conformational changes in the protein during ligand binding [123–128]. One of the methods of studying this phenomenon is molecular dynamics. If the X-ray structure of the appropriate ligand-protein complex is not available, the poses selected in the docking can be used to set up MD simulations [85].

In drug design, molecular dynamics simulations allow the study of a wide variety of processes, including changes in protein conformation caused by ligand binding or unbinding and by the flexibility of different protein regions. MD simulations allow us to predict or explain the effects of the mutations on a protein's structure and function [22,23,25]. Simulation-based methods also enable more accurate estimation of ligand binding affinity than docking [129–131]. Of course, molecular dynamics also have their limitations. They mainly result from approximations in force fields, the need for appropriate parameterization of ligands, the lack of taking into account electronic effects such as polarization, formation or breaking of covalent bonds, and the time scales of the studied phenomenon [129,131]. Advances in algorithms, improvements of computer hardware, especially the use of graphics processing units (GPUs) in MD simulations along with enhanced sampling techniques [132–135] allow simulations to study events taking place over a longer time scale [131,136].

## 4. Summary: Perspectives of CYP1B1 Studies in Drug Design

Considering the association of CYP1B1 with metabolic diseases, carcinogenesis, and eye diseases, the interaction of this enzyme with substrates/inhibitors is an important objective for studies with the use of the latest computational methods. Recent studies indicate the flexibility of ligand as a feature that helps in fitting to the enzyme cavity. In addition, the shape, electrostatic environment, and malleability of enzyme active site significantly determine ligand-enzyme interactions. Molecular docking and, in particular, molecular dynamics simulations, allow extending our knowledge of the enzyme's molecular structure, which determines its function. Knowing the molecular mechanisms which drive the processes essential for organism functioning will enable us to cure the diseases associated with metabolic disorders and redox imbalance. The design of efficient and safe CYP1B1 inhibitors is still a developing area of research.

**Author Contributions:** R.M., Z.D. contributed equally to the manuscript development and preparation. Both authors have read and agreed to the published version of the manuscript.

**Funding:** This research received no external funding.

**Institutional Review Board Statement:** Not applicable.

**Informed Consent Statement:** Not applicable.

**Data Availability Statement:** Not applicable.

**Acknowledgments:** This study was supported by grants from Poznań University of Medical Sciences and Nicolaus Copernicus University in Toruń, Ludwik Rydygier Collegium Medicum, Statutory Research no. PDB WF411.

**Conflicts of Interest:** The authors declare no conflict of interest.

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
