# Peer review of "New Perspectives of CYP1B1 Inhibitors in the Light of Molecular Studies"

_processes, doi:10.3390/pr9050817_

Round 1

Reviewer 1 Report

The manuscript entitled „New perspectives of CYP1B1 inhibitors in the light of molecular studies” is the review work, which revealed the role of this specific isoenzyme in the pathogenesis of some diseases as well as in the cancer chemoprevention and therapy. The authors also added the results related to CYP1B1 molecular structure in respect for the design of the effective enzyme inhibitors, which have the chance to be the future therapeutics.

In my opinion the presented manuscript is a good piece of work. It considered several aspects in the relation to several diseases including eye pathology, the oxygen supply as well as cardiovascular diseases. In the field CYP1B1 mediated metabolism authors considered the disorders of endogenous metabolism with CYP1B1disruption leading to the cancer prevention and therapy. Among others Authors cited their works in respect to CYP1B1 mediated chemoprevention agents, considering inhibitor effects on the conformational changes in the enzyme structure [60]. There were considered that the co-administration of the anticancer agents and CYP1B1 inhibitors would decrease drug resistance making the possible better outcome of anticancer therapy. Authors also underlined that the high CYP1B1 activity in cancer cells allows for the designing of prodrugs activated by this enzyme directly in tumor. The molecular aspects of CYP1B1 studies in respect to the structure of the active centre and to the design of the selective enzyme inhibitors took up the significant part of this work. There were also considered the references to the authors original works [62, 81 and 88].

I would like to underline the good aspects of this work.

  1. A well chosen particular subjects of the review.
  2. Very good description of CYP1B1 inhibitors, paragraph 4.2. This paragraph is long, but it is good for the reading. All have to agree that the statement: “CYP1B1 with reduced enzyme function are responsible for better response to anticancer treatment” should be confirmed, but it is the high potential of CYP1B1 action.

The week aspect of this work

                Page 2, line 64-66

I cannot agree that CYP1A1 and CYP1A2 participate mainly in detoxification of xenobiotics. There are also very often occur the metabolic activation with P4501A1 and 1A2 to carcinogenic compounds.

Page 5,

Paragraph 4.1. seems to be long. Particularly, there are not clear the lines 300-315

                On several pages

There are the lack of the explanation of some abbreviations.

Generally, they are in the text, but sometimes when they are repeated after several pages it is difficult to find once more, for example PCG, AOEG and POAG on page 11 (they are explained on page 2), what means BMP6 on page 2? line 93 and compound 49a on the description of Figure 2.

                General aspect

There are three figures illustrated the structures in the active centre of the enzyme. However, there is the lack of some graphs or schemes, which would illustrate other parts of the text to make it easier to understand

Author Response

We would like to thank the reviewer for valuable comments, which helped us to improve our manuscript significantly.

  1. We reformulated the sentence ( Page 2, line 64-66). We hope that now it is more clear.
  2. The paragraph concerning ancestral structure of CYP1B1 we changed in order to obtain more comprehensible form by eliminating some details.
  3. Paragraph 4.1 as too long was divided to two parts: 4.1. Crystal family CYP1 structures and 4.2. CYP1B1 structure studies
  4. To improve comprehensibility of the text, we repeated the explanation of abbreviations on page 11 and introduced the abbreviation for BMP6. The compound 49a is named as a benzo[h]chromone derivative. The abbreviation AOEG appeared to be a mistake.
  5. We enriched the manuscript with the schemes illustrating the contribution of CYP1B1 in pathogenesis of some diseases and its role in chemoprevention and therapy of cancer (Figure 1 and 2). We introduced the Figure 6 with core structures of studied inhibitors of CYP1B1 activity.

Reviewer 2 Report

The manuscript entitled “New Perspectives of CYP1B1 Inhibitors in The Light of Molecular Studies” provided a review for therapeutics targeting CYP1B1. This review covers a wide range of topics related to CYP1B1 including the roles of CYP1B1 in diseases, structural properties on ligan recognition of CYP1B1, and recent several trials for finding new inhibitors with in silico methods. I think this provides great insights for readers who are interested in drug discovery targeting CYP family.

I would like to raise some suggestions to improve the quality of this review article.

[1] More detailed information about computational methodologies should be provided. Although the recent advances in computational technologies have increased reliability of in silico methods, the reliability largely depends on calculation conditions. For example, although Ref [24] at the Line 83 in Page 2, which is published in 2006, provided great insight into the effects of mutations on three-dimensional structure of CYP1B1 by using molecular dynamics method, the simulations reported in this paper are well behind current studies. Today, it is considered that 30-ns time courses of the united atom models may not be enough to investigate conformational diversity derived from mutations. In the same way, the reliability of docking methods also largely depends on their conditions.

              Reliability of previous computational studies, both the molecular dynamics and molecular docking methods, should be commented.

[2] At the line 242 in Page 6, what is “new computational methods” should be explained briefly.

[3] In the paragraph began at the line 277 in Page 7, the first sentence is “More recently, the new structure of CYP1B1 with inhibitor having azide group as a ligand (PDB id: 6IQ5) was crystallized [76]” but the Line 283 says “the site of azide group interactions with CYP1B1 was identified by molecular docking;”. I am confused by these descriptions. I think the interactions of azide group can be identified by the crystal structure. Why molecular docking was needed to identify the interactions and what is new finding in the docking study should be paraphrased.

[4] In the subsection “4.2. Design of selective CYP1B1 inhibitors”, interactions of several candidate compounds are discussed. Adding figures or schemes showing chemical structures would help readers.

[5] Definition of abbreviations should be checked again. For example, what ROS stands for is not defined. The abbreviation AhR is defined twice.

[6] At the Line 384 in Page 10, a white space should be inserted between “π-π” and “stacking”.

[7] Font styles of the main text should keep consistency.

Author Response

We appreciate the valuable suggestions of the reviewer, which helped us to improve the quality of our manuscript.

  1. We provided more information about computational techniques; particularly focussing on current studies.

We added a subsection 3.5 devoted to the discussion of currently used in silico methods.

  1. We reformulated the sentence (page 6, line 242) referring to the new subsection 3.5.
  2. Why molecular docking was needed to identify the interactions and what is new finding in the docking study should be paraphrased.

In the manuscript, we expanded on the purpose and the results of docking analysis of Kubo et al. by providing the paragraph:

X ray crystal structure allowed to confirm the location of the inhibitor molecule in the CYP1B1 cavity, while the positioning of terminal nitrogen atom could not be determined due to low resolution. The authors (Kubo et a., 2019) perfomed docking analysis using Surflex Dock program on Sybyl X2 software and found that azide group is located in the hydrophobic pocket constructed by Thr334, Val395 and Thr510. The results exhibit the role of hydrophobic interactions affecting the binding affinity and inhibitory potency of CYP1B1 inhibitors.

  1. We included a figure with the core structures of CYP1B1 inhibitors being discussed in the manuscript (Figure 6). We added the schemes illustrating multiple functions of CYP1B1 and the role of CYP1B1 in pathogenesis of diseases (Figure 1) and the role of CYP1B1 in chemoprevention and therapy of cancer (Figure 2).

5, 6, and 7. All indicated mistakes are corrected.

Round 2

Reviewer 2 Report

Authors have addressed all the points in the previous review.